# Changes in Water and Sediment Processes in the Yellow River and Their Responses to Ecological Protection during the Last Six Decades

**Suiji Wang** [1,2,*] and **Xumin Wang** [2,3]

1   Key Laboratory of Water Cycle and Related Land Surface Processes, Institute of Geographic Sciences and Natural Resources Research, Chinese Academy of Sciences, Beijing 100101, China
2   College of Resources and Environment, University of Chinese Academy of Sciences, Beijing 100049, China; wangxumin22@mails.ucas.ac.cn
3   Xinjiang Institute of Ecology and Geography, Chinese Academy of Sciences, Urumqi 830011, China
*   Correspondence: wangsj@igsnrr.ac.cn

**Abstract:** The variation of river hydrologic process can reflect the impact of not only natural factors, but also human activities. The purpose of this study is to reveal the change in the hydrologic regime of the Yellow River and its response to ecological protection. Based on the daily water and sediment observation data of representative gauging stations of the Yellow River, we analyzed the variation of the annual and monthly runoff and suspended sediment load (SSL), as well as monthly mean runoff, suspended sediment transport rate (SSTR), sediment inflow coefficient, and hydrological regime in a decadal average of the gauging stations during the period of 1960–2019. The results showed that the variation of annual runoff and SSL, as well as the monthly mean runoff and SSTR in a decadal average, had a significant decreasing trend in the 1960s–1990s, which was mainly in response to the gradual implementation of ecological protection measures such as afforestation, terrace construction, check dam construction, etc., in the basin. In 2000s and 2010s, the annual runoff increased, while the SSL increased slightly. This was a response to the implementation of new river management measures such as ensuring the ecological water demand of the lower reaches and scouring the riverbed by manually regulated water discharged from the Xiaolangdi Reservoir. At the same time, the monthly mean runoff and SSTR for the flood season (July–October) decreased remarkably while the process curve of the monthly mean discharge and sediment concentration changed from a clockwise loop to a counterclockwise loop in the river reach below the Xiaolangdi dam. This was a comprehensive response to the environmental protection measures in the Yellow River basin, in which the construction and operation of the Xiaolangdi Reservoir played a key role. This study can provide reference for river basin management.

**Keywords:** runoff; sediment load; hydrologic regime; change trend; influence factors; Yellow River

## 1. Introduction

River basin is the basic geomorphic unit of surface runoff and sediment production, and river is an important channel for transporting water and sediment. Large rivers on earth that eventually flow into the sea are the key channels to provide water and sediment fluxes from land to sea. Therefore, the amount of water and sediment transported by the river and its changes not only affect the environmental changes in the estuary area, but also reflect the changes in the conditions of water and sediment production in the basin. Generally, when the sedimentation in the lower reaches of a river basin or in the estuary area increases, the denudation in the sediment yield area of the basin also increases, and vice versa.

At least a century ago, most of the rivers in the world were basically in the stage of natural evolution. The water and sediment processes of these rivers were mainly affected

by climate factors such as precipitation and evaporation. In recent decades, the evolution of these rivers has superimposed more and more intense impacts of human activities, and the water and sediment changes in many rivers, especially in arid and semi-arid regions of the world, are mainly controlled by human activities (e.g., [1–8]). Therefore, the river water and sediment tend to decrease in the last decades, while the global flux of fluvial sediment to the oceans has reduced significantly compared with the previous period [9,10]. It is estimated that under the effect of human activities superimposed on climate change, the global land–sea fluvial sediment flux has decreased from about 19 billion tons per year [11] to about 13 billion tons per year, with a decrease of about 30% [12,13].

Although fluvial sediment is composed of suspended and bed loads, the suspended load transported by many rivers accounts for about 90% of the total sediment loads (e.g., [14–17]). Therefore, the impact of human activities on river sediment is mainly reflected in the impact on its suspended sediment load. The Yellow River, once known for its extremely high sediment concentration and as the largest carrier of fluvial sediment in the world, has experienced a dramatic reduction in its runoff and sediment transport in recent decades [2,5,8,18–24]. For instance, the annual suspended sediment transported by the Yellow River to the Bohai Sea decreased from 1.25 billion tons per year during the time period of 1952–1968 to 132 million tons per year during the period of 1997–2018 [13], with a decrease of up to 89.4%. Many research results show that this is mainly affected by the implementation of long-term comprehensive water and soil conservation and ecological protection measures in the Yellow River basin, of which vegetation restoration and water conservancy project construction play a pivotal role (e.g., [5,8,13]).

Despite the fact that there have been many studies on the interannual and decadal changes in runoff and sediment in the Yellow River basin, there are still insufficient studies on the seasonal changes in water and sediment and the adjustment of water and sediment relationship. The purpose of this study is to (1) compare the annual and monthly variation trends and differences of runoff and suspended sediment transport in the middle and lower reaches of the Yellow River; (2) analyze the characteristics of monthly average sediment inflow coefficient, the change in relationship between monthly mean discharge and sediment concentration in different decades; and (3) reveal the influence mechanism of environmental protection and dam construction in the Yellow River basin on the change in water and sediment volume and water and sediment relationship of the Yellow River. This work will help to further understand the change mechanism of water and sediment in the Yellow River and provide reference for the effective protection of ecological environment in river basins including the Yellow River basin.

## 2. Study Area and Dataset

### 2.1. Study Area

The Yellow River, which originates from the Qinghai–Tibetan Plateau, traverses the Loess Plateau and the North China Plain, finally flows into the Bohai Sea (Figure 1), has a total length of 5464 km and a total drainage area of 0.75 million km$^2$ [4,25]. The river basin is commonly divided into the upper (above Toudaoguai), middle (from Toudaoguai to Xiaolangdi), and lower (below Xiaolangdi) reaches with a channel length of 3472, 1206, and 786 km and a drainage area of 0.39, 0.34, and 0.02 million km$^2$, respectively [19,26]. The annual mean runoff produced in the three reaches accounts for 54%, 36%, and 10% of the total runoff of the Yellow River basin, while the annual mean suspended sediment yield from the river reaches accounts for 10%, 88%, and 2% of the total annual mean suspended sediment load.

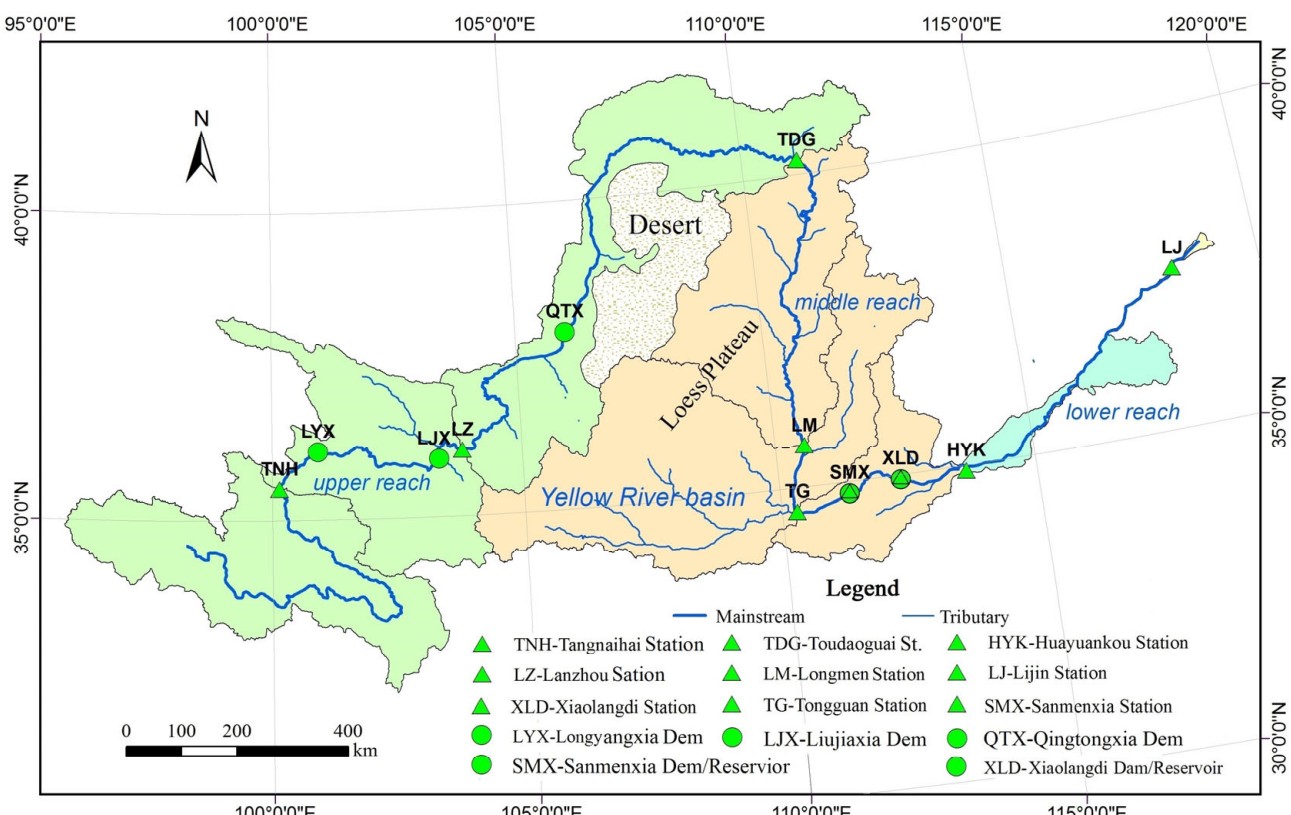

**Figure 1.** The watershed and locations of major hydrological stations and large reservoirs of the Yellow River.

The river catchment above the Tangnaihai (TNH) hydrological station, commonly known as the source of the Yellow River (Figure 1), is located on the Qinghai–Tibet Plateau, with a watershed area of 0.122 million km$^2$, annual mean runoff of 200.02 billion m$^3$, and annual mean SSL of 11.87 million tons, which account for 16.3%, 50.8%, and 1.1% of the total area, mean runoff and SSL of the entire river basin, respectively. The river reach from Tangnaihai to Xiaolangdi (XLD) hydrological stations is located on the Loess Plateau, with a watershed area of 0.57 million km$^2$, annual mean runoff of 302.21 billion m$^3$ and SSL of 960 million tuns, accounting for 70.0%, 39.2% and 88.9% of that for the total river basin, respectively.

Most areas of the Yellow River basin are located in arid and semi-arid climate regions with a mean annual temperature of 8–14 °C and a mean annual precipitation of 478 mm [19]. Although the mean annual precipitation for the upper, middle, and lower reaches is 368, 530, and 670 mm, respectively [20], the most runoff of the Yellow River originates from the upper reach, while the suspended sediment mainly yields from the middle reach [25]. The Yellow River basin contains 12.6 million hectares of farmland and more than 40% of the farmland is irrigated using the water from the Yellow River [27] (Xia et al., 2002).

As a key engineering measure for ecological protection, the Longyangxia and Liujiaxia reservoirs located in the upper reach of the Yellow River were completed in 1968 and 1986, respectively (Figure 1). The Sanmenxia and Xiaolangdi reservoirs located near the outlet of the middle reach of the Yellow River were completed in 1961 and 2001, respectively. The operation of these reservoirs can make relatively efficient use of the water resources, provide ecological water demand and agricultural water in the dry seasons, and prevent floods in the rainy seasons. The water stored in these reservoirs can produce clean energy such as hydropower, which not only ensures the local residents' demand for electricity in industrial and agricultural production and daily life, but also reduces the consumption of fossil energy.

## 2.2. Dataset and Method

At the outlets of the middle and lower reaches of the Yellow River, control hydrological gauging stations of Xiaolangdi and Lijin were set up with a controlled drainage area of 0.6904 and 0.738 million km$^2$, respectively. These two hydrological stations are key stations for understanding the changes in water and sediment in the middle reach and the entire basin of the Yellow River, respectively. In addition to the two hydrological stations mentioned above, this study also involves the Tongguan and Sanmenxia gauging stations in the middle reach and the Huayuankou gauging station in the lower reach of the Yellow River, which control a watershed area of 0.682, 0.688, and 0.730 million km$^2$, respectively. These hydrological stations are located upstream of Sanmenxia Reservoir, between Sanmenxia and Xiaolangdi Reservoirs, and in the alluvial river reach downstream of Xiaolangdi Reservoir, respectively. The differences in water and sediment processes between the stations can reveal the impact of reservoir construction and operation on hydrological mechanisms.

This paper uses the annual runoff and suspended sediment load (SSL) of the Tongguan, Sanmenxia, and Lijin hydrological stations gauged from 1950 to 2021 to investigate their interannual and interdecadal variation trends and analyze the differences between the stations (Figures 2 and 3). Based on the daily observation data of Sanmenxia, Xiaolangdi, Huayuankou, and Lijin hydrological stations during 1960–2019, the focus of this article is to analyze the monthly variation of relevant hydrological parameters and their mean monthly variation in terms of decadal average. The monthly mean values of runoff, SSL, coefficient of sediment inflow, discharge, and suspended sediment concentration (SSC) for the four stations were calculated from the basic data. Based on the above calculation results, the decadal mean values of the corresponding parameters in each month for the hydrological gauging stations were also calculated. All of the daily original data and annual data were obtained from the Interior Reports of Yellow River Water Conservancy Commission [28].

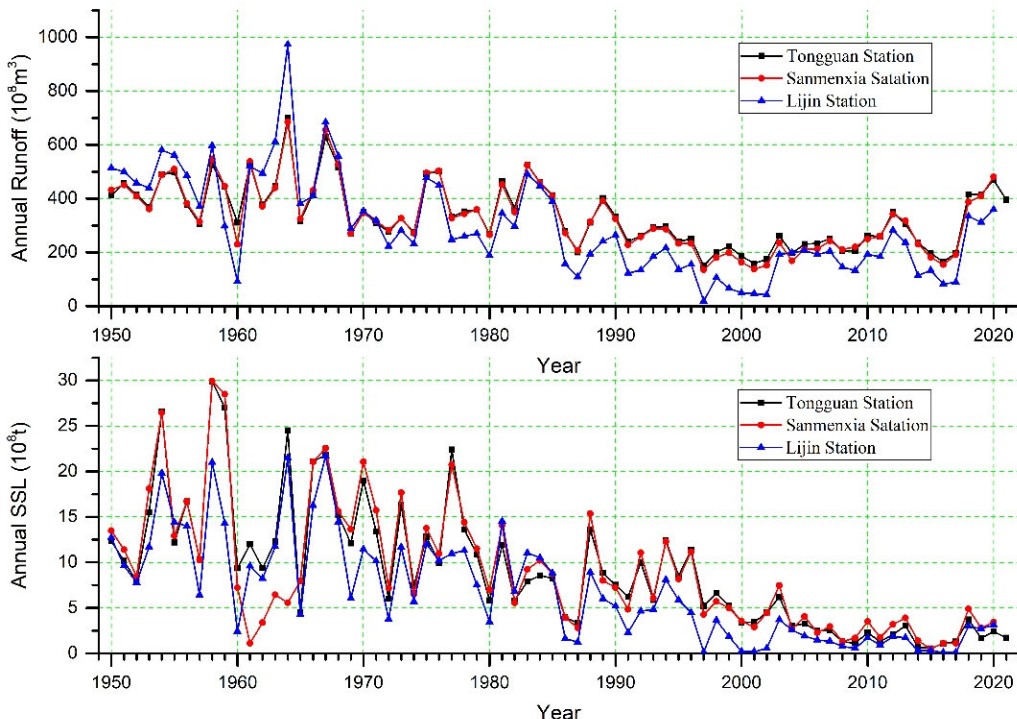

**Figure 2.** Annual variations of runoff and suspended sediment load (SSL) at the Tongguan, Sanmenxia, and Lijin gauging stations of the Yellow River.

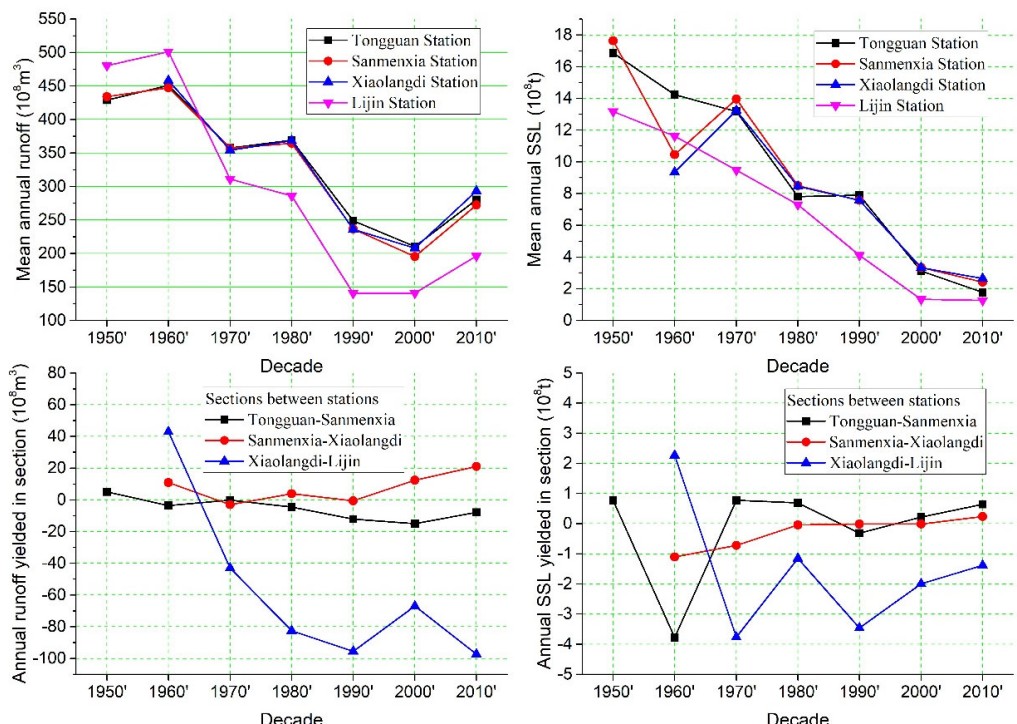

**Figure 3.** Annual mean runoff and suspended sediment load (SSL) in decadal average at the Tongguan, Sanmenxia, Xiaolangdi, and Lijin gauging stations and their net yield in the different sections of the Yellow River during the last decades.

## 3. Results

### 3.1. Annual Variations of Runoff and SSL

The annual runoff and SSL during the time period from 1950 to 2021 were examined at the Tongguan, Sanmenxia, and Lijin stations in the Yellow River (Figure 2). The runoff fluctuated with a low amplitude and had no significant trend of change in the 1950s, while it fluctuated greatly in the 1950s and its maximum value appeared in 1964. The runoff showed a trend of decreasing with minor fluctuations during 1964–1997 and an increasing trend in the time period of 1997–2021. The annual SSL at the three stations presented an overall decreasing trend and their interannual fluctuations also decreased with time (Figure 2). Compared with the three hydrological stations, the values and variation characteristics of the Tongguan and Sanmenxia stations were very similar in terms of annual runoff and SSL.

The mean annual runoff and SSL at four gauging stations (Tongguan, Sanmenxia, Xiaolangdi, and Ljin) as well as the net water and sediment yield in three sections (Tongguan–Sanmenxia, Sanmenxia–Xiaolangdi, and Xiaolangdi–Ljin) in different decades are shown in Figure 3.

The variation trends in annual mean runoff of the four hydrological stations in different decades are generally similar, with Tongguan, Sanmenxia, and Xiaolangdi gauging stations displaying very similar trends. In comparison, the runoff difference between Lijin and other hydrological stations is relatively large, with the largest among all four hydrological stations in the 1950s and 1960s, and the smallest in the subsequent decades (Figure 3). For the seven decades, the largest annual mean runoff occurred in the 1960s, while the smallest occurred in the 2000s (1990s for the Lijin Station), with the latter accounting for only 46.7%, 43.7%, 45.4%, and 28.0% of the former.

The mean annual SSL of these four hydrological stations in the different decades has a monotonic decreasing trend with time changes (with the exception of Sanmenxia and Xiaolangdi in the 1960s) (Figure 3). Except for Xiaolangdi, the minimum values of the other three hydrological stations appeared in the 2010s and the maximum values appeared in the 1950s, with the former accounting for only 10.5%, 13.7%, and 9.6% of the latter. For all four

hydrological stations, the sudden decrease in mean annual SSL mainly began in the 1970s, with decreases of 86.5%, 82.7%, 80.0%, and 86.6% compared to those in the 2010s.

Net yield in mean annual runoff for the section of Tongguan–Sanmenxia showed a slightly decreasing trend (from 0.5 billion $m^3$ $yr^{-1}$ in the 1960s to −15 billion $m^3$ $yr^{-1}$ in the 2000s, then it slightly increased to −7.9 billion $m^3$ $yr^{-1}$ in the 2010s), while net yield in mean annual suspended sediment in the section was only tens of millions of tons (46.3 million tons $yr^{-1}$ in average), except for −380 million tons $yr^{-1}$ in the 1970s (Figure 3). The net yield in mean annual runoff for the section of Sanmenxia–Xiaolangdi fluctuated with small ranges in the 1960–1980s while it increased slightly since the 1990s. The net yield of mean annual SSL for the section showed an increasing trend from −110 to 20 million tons $yr^{-1}$ in the decades from the 1960s to 2010s. For the section of Xiaolangdi–Lijin, the net yield of mean annual runoff showed a decreasing trend from 4.2 to −9.7 billion $m^3$ $yr^{-1}$ except for in the 2000s. The net yield of SSL for the section was 0.23 billion tons $yr^{-1}$ in the 1960s, and then it fluctuated in the range between −0.38 and −0.12 billion tons $yr^{-1}$.

*3.2. Monthly Variation of Runoff and SSL*

The variation of monthly runoff and SSL at the Sanmenxia and Lijin gauging stations and their net yield in the section between the stations of the Yellow River during the time period from January 1960 to December 2020 (the month order is from 1 to 732) are shown in Figure 4. The maximum monthly runoff decreased in the decades from the 1960s to the 1990s and then increased in small ranges for the two stations. The minimum and maximum monthly net runoff yield in the section between Sanmenxia to Lijin was in the 1960's and 2010's, respectively. In comparison, the maximum monthly SSL per year increased in the 1960s, fluctuated significantly in the 1970s and 1980s, and decreased significantly since the 1990s. The monthly net suspended sediment yield in the section was generally the smallest and dominated by negative values (scouring) in the 1960s, and in the 1980s, positive and negative values alternated (scouring and silting). Since then, the maximum monthly net sediment yield in various decades has shown a downward trend.

*3.3. Change in Monthly Runoff and SSL in Decadal Average*

The changes in monthly runoff in decadal average at the Sanmenxia, Xiaolangdi, Huayuankou, and Lijin hydrological stations in the six decades from 1960 to 2019 are shown in Figure 5. The monthly average runoff variation curve at the Sanmenxia hydrological station (Figure 5a) shows that in the decades before 1990, the three high values of the runoff occurred in the flood season (from July to October), while the three low values occurred in winter (January, February, and December). In the three decades from 1990, the mean monthly runoff in the flood season significantly decreased, and the monthly average runoff distribution curve in the flood season changed from a convex peak shape in previous decades to a peak shape, that is, the mean monthly runoff value in August or September was the maximum value in each decade, while the values in other months of the flood season significantly decreased compared to those of the previous decades. Especially in the 2000s and 2010s, the maximum mean monthly runoff during the flood season was less than half of that in each decade of the 20th century. The minimum and sub-minimum of the mean monthly runoff occurred in January and May, which was 840 and 989 million $m^3$ in the 2000s and 1.22 and 1.30 million $m^3$ in the 2010s, respectively. The total runoff during the flood season was 8.64 billion $m^3$ and 14.22 billion $m^3$ in the recent two decades, respectively, accounting for only 34.4% and 56.6% of the total runoff during the flood season in the 1960s.

The distribution curve of the monthly runoff in decadal average at Xiaolangdi gauging station was very similar to that of the Sanmenxia gauging station over the last four decades of the 20th century (Figure 5b), but a significant difference between distribution curves of the stations was in the 2000s and 2010s, mainly manifested in the fact that the maximum or sub-maximum of the monthly runoff for Xiaolangdi station occurred in June (non-flood season). The total runoff during the flood season in the last two decades was 6.91 billion $m^3$

and 11.39 billion m³, respectively, accounting for only 26.9% and 44.3% of that during the flood season in the 1960s.

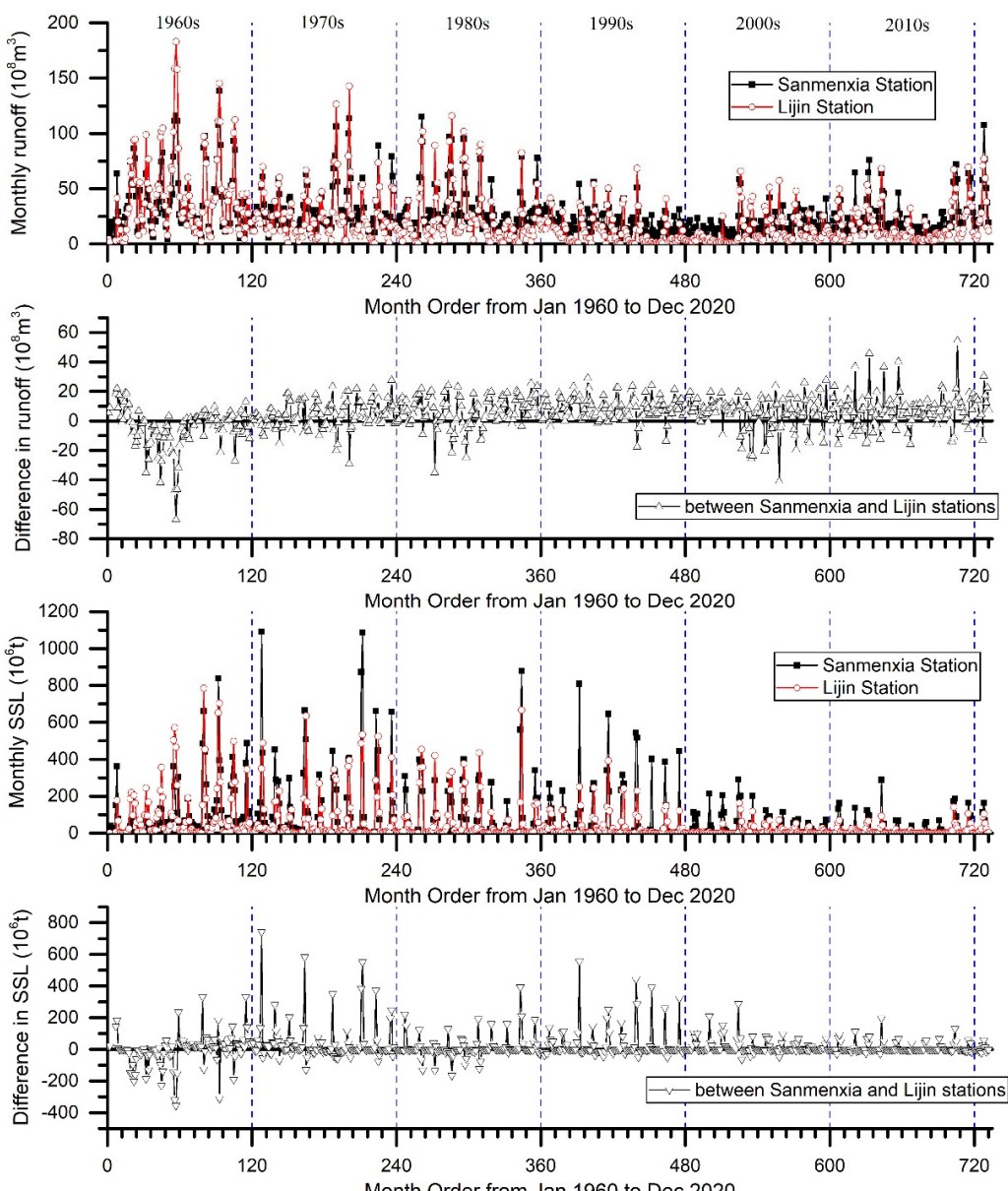

**Figure 4.** Variation of monthly runoff and SSL at the Sanmenxia and Lijin gauging stations and their net yield in the section between the two stations of the Yellow River during the time period from January 1960 to December 2020 (the serial number of months is from 1 to 732).

The distribution curve of the monthly runoff in the decadal average of the Huayuankou gauging station (Figure 5c) in the three decades before 1990 is also roughly similar to that at Sanmenxia and Xiaolangdi stations. In the 2000s and 2010s, the peak shape changed from a convex peak in the previous decades to a sharp peak, and the peak tip gradually shifted from October to July, with the peak tip in the 2000s even appearing in June, which belongs to the non-flood season. The total runoff in the flood season of the last two decades was 8.6 billion m³ and 1.24 billion m³, respectively, accounting for only 29.9% and 43.1% of that in the 1960s.

The distribution curve of the monthly runoff in the decadal average (Figure 5d) at the Lijin gauging station is roughly similar to that of the above three stations in the three decades before 1990. In the 2000s, the peak shape in the flood season changed from a convex

peak for the previous decades to a double peak, while in the 2010s, it showed a sharp peak. The peak tip also gradually shifted from October to July, but all peaks were distributed in the flood season. The total runoff during the flood season in the last two decades was 7.36 billion m$^3$ and 10.36 billion m$^3$, respectively, accounting for only 25.3% and 35.5% of that of the 1960s, which was the largest decline among the four gauging stations.

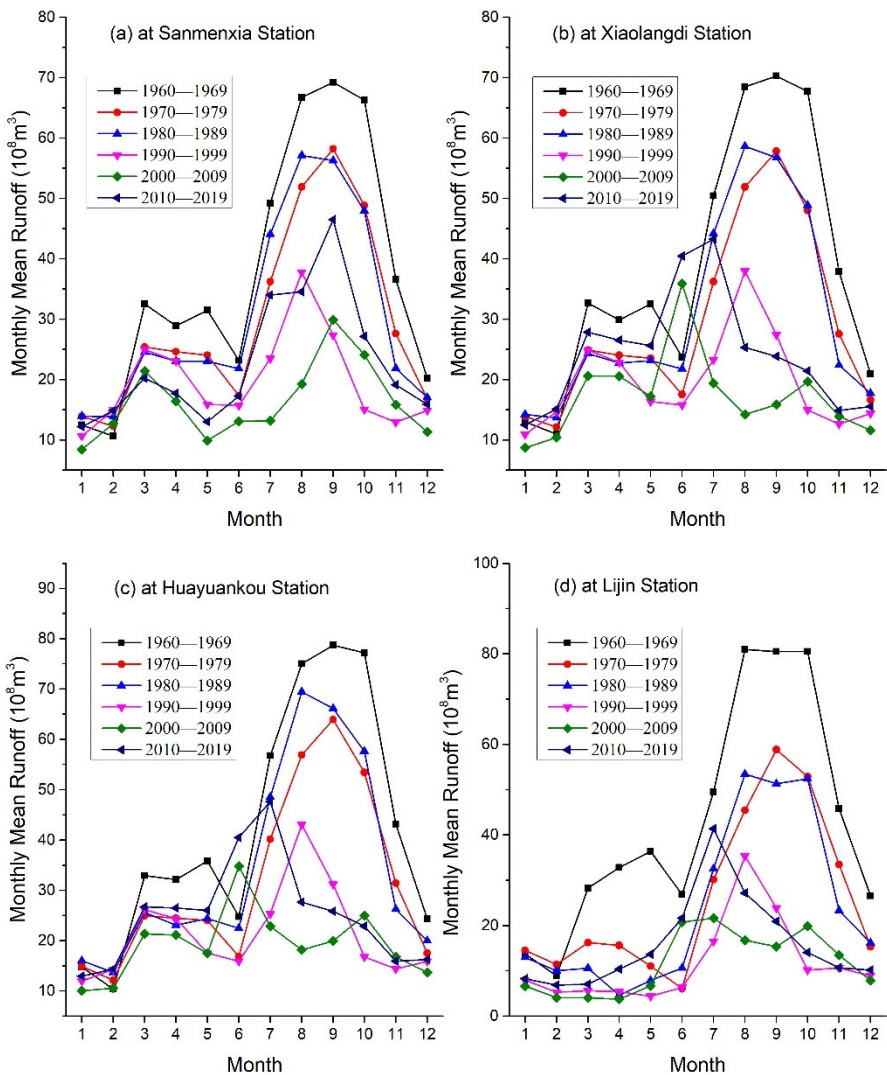

**Figure 5.** Distribution curves of the monthly runoff in decadal average at the Sanmenxia, Xiaolangdi, Huayunakou, and Lijin gauging stations of the Yellow River.

There are significant seasonal differences in water and sediment changes in the Yellow River basin (Figure 6). During the flood seasons, the monthly mean runoff at the hydrological stations of the middle and lower reaches of the Yellow River presented a synchronous significant decrease trend with the passage of time, and reached the minimum value in the 2000s, only increasing in the 1970s and 2020s compared to the previous period (Figure 6a). The SSTR during the flood season has generally shown a significant decrease trend since the 1970s. Except for the small decrease in the 2010s, the decrease in other time periods was significant (Figure 6b). In comparison, the monthly mean runoff in non-flood seasons decreased generally in the last century, mainly increasing in this century (Figure 6c), while the monthly mean SSTR in non-flood seasons showed a decreasing trend except in the 1990s, with a smaller extent compared to that of the flood seasons (Figure 6d).

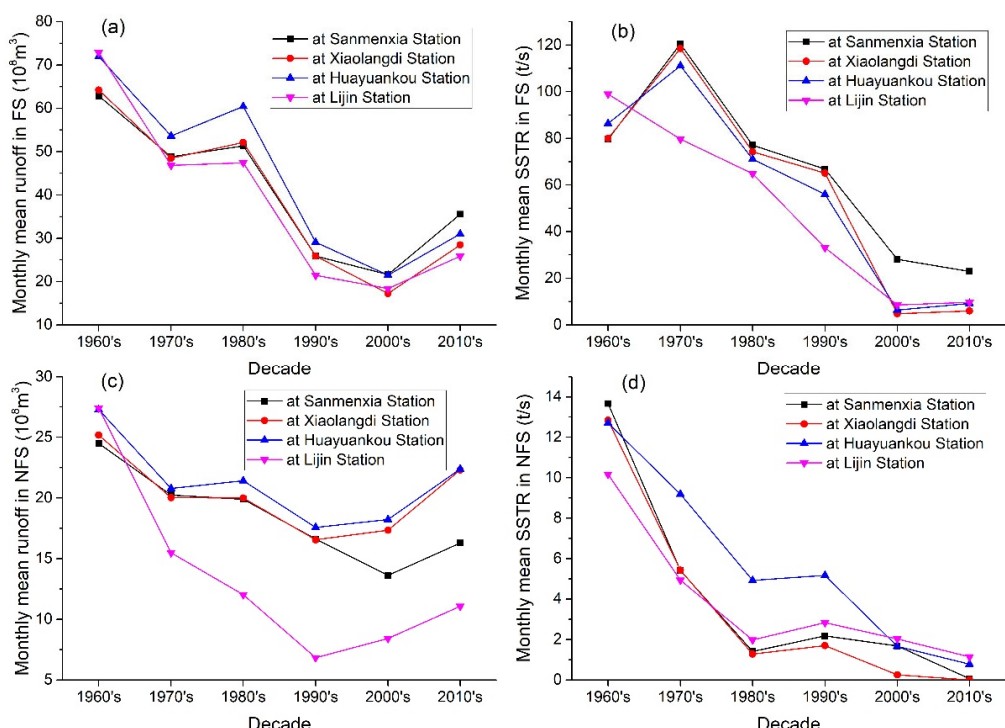

**Figure 6.** Changes in the monthly mean runoff and SSTR in decadal average in flood season (FS) and non-flood season (NFL) at the Sanmenxia, Xiaolangdi, Huayunakou, and Lijin gauging stations of the Yellow River. (**a**) monthly mean runoff in flood season, (**b**) monthly mean SSTR in flood season, (**c**) monthly mean runoff in non-flood seasons, and (**b**) monthly mean SSTR in non-flood season.

### 3.4. Change in Monthly SSTR in Decadal Average

The change in the monthly suspended sediment transport rate (SSTR) in decadal average at the Sanmenxia, Xiaolangdi, Huayuankou, and Lijin hydrological stations during the six decades from 1960 to 2019 is shown in Figure 7. The monthly variation curve of the SSTR at the Sanmenxia station (Figure 7a) showed that in the four decades before 2000, the high values occurred in the three months from July to September (belonging to the flood season), with the largest in the 1970s and similar in the other three decades. The low values appeared in winter and early spring. In the 2000s and 2010s, the monthly mean SSTR during the flood season significantly decreased, and its distribution curve shape in the flood season changed from the previous high peak to a gentle peak. The mean SSTR during the flood season in the 2000s and 2010s only accounted for 35.3% and 28.8% of that in the 1960s.

For the Xiaolangdi hydrological station, the distribution curve of the monthly mean SSTR in the four decades of the 20th century (Figure 7b) was almost the same as that of the Sanmenxia station, but in the 2000s and 2010s, it was quite different from that of the Sanmenxia station in the same period. The main differences are that the monthly mean SSTR during the flood season significantly decreased, and there was no significant difference compared to other months. The mean SSTR during the flood season in these two decades only accounted for 6.0% and 7.5% of that during the same period in the 1960s.

The distribution curves of the monthly mean SSTR (Figure 7c,d) at the Huayuankou and Lijin gauging stations are generally similar to those at the Xiaolangdi station, with the difference that the monthly mean SSTR during the flood season gradually decreased over time, which was particularly evident at the Lijin station. Therefore, the value for the two hydrological stations during the flood season in the 2000s were only 7.4% and 8.6% of those in the 1960s, while those in the 1990s were only 10.7% and 9.8%, respectively.

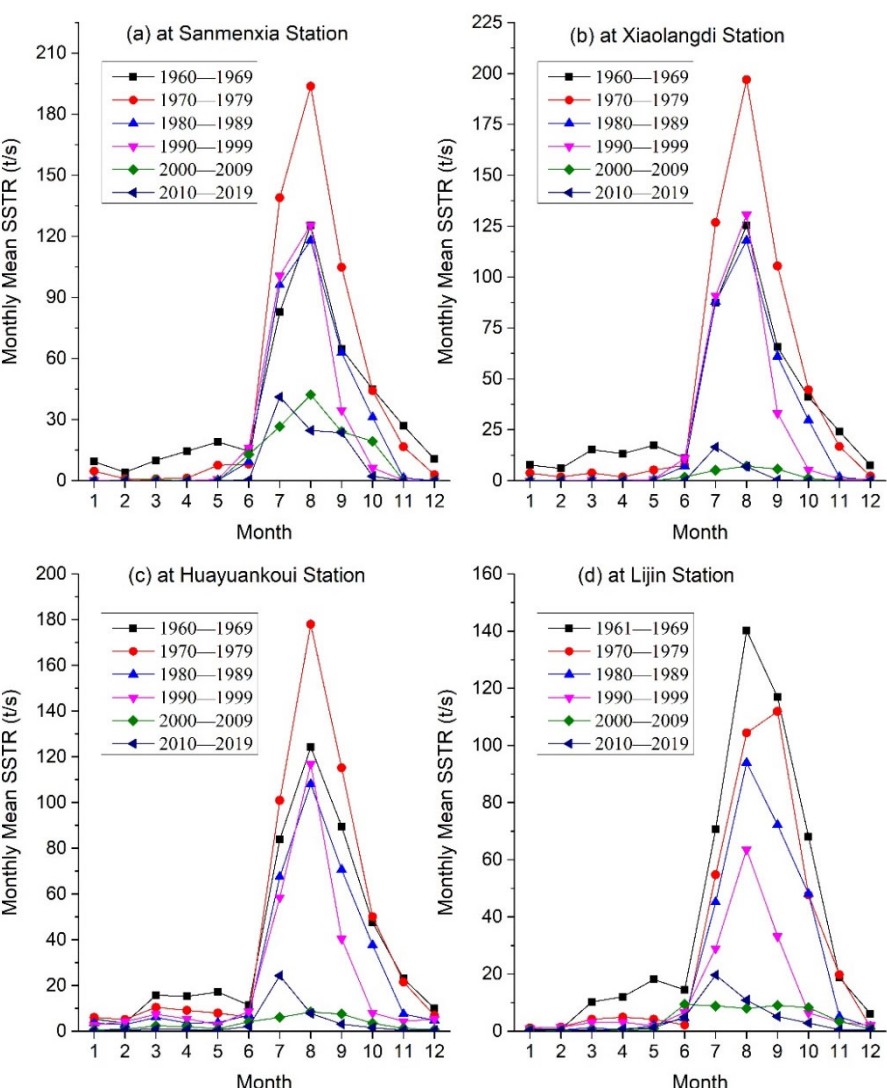

**Figure 7.** Monthly mean suspended sediment transport rate (SSTR) in decadal average at the Sanmenxia, Xiaolangdi, Huayunakou, and Lijin stations of the Yellow River.

### 3.5. Change in Monthly Sediment Inflow Coefficient

The sediment inflow coefficient is defined as the ratio of sediment concentration to water discharge, and its unit is kg·s/m$^6$. As pointed out by [29], it can indicate the sediment concentration per unit discharge or per unit flow power, implying the ratio of the sediment concentration that reflects the scouring and silting capacity of the river to the critical sediment concentration. The sediment inflow coefficient can be used as a judgment for river scouring and silting, and is also a key parameter in the formula in terms of non-equilibrium transport of sediment. Therefore, analyzing the temporal and spatial changes in the sediment inflow coefficient is of great significance for understanding the changes in river flow and sediment, as well as the changes in riverbed erosion and deposition.

The variation trend of monthly sediment inflow coefficient in decadal average at the Sanmenxia, Xiaolangdi, Huayuankou, and Lijin hydrological stations in the six decades from 1960 to 2019 is shown in Figure 8. Its monthly variation curve at the Sanmenxia station (Figure 8a) shows that in the 1960s, the monthly fluctuation was relatively low, with high values occurring in winter with low flow and low sediment concentration and summer with high flow and high sediment concentration. Moreover, the curve crest in winter is significantly higher than that in autumn, which is mainly the result of a large amount of sediment interception during the flood season and sediment discharge during the non-flood season of the Sanmenxia Reservoir.

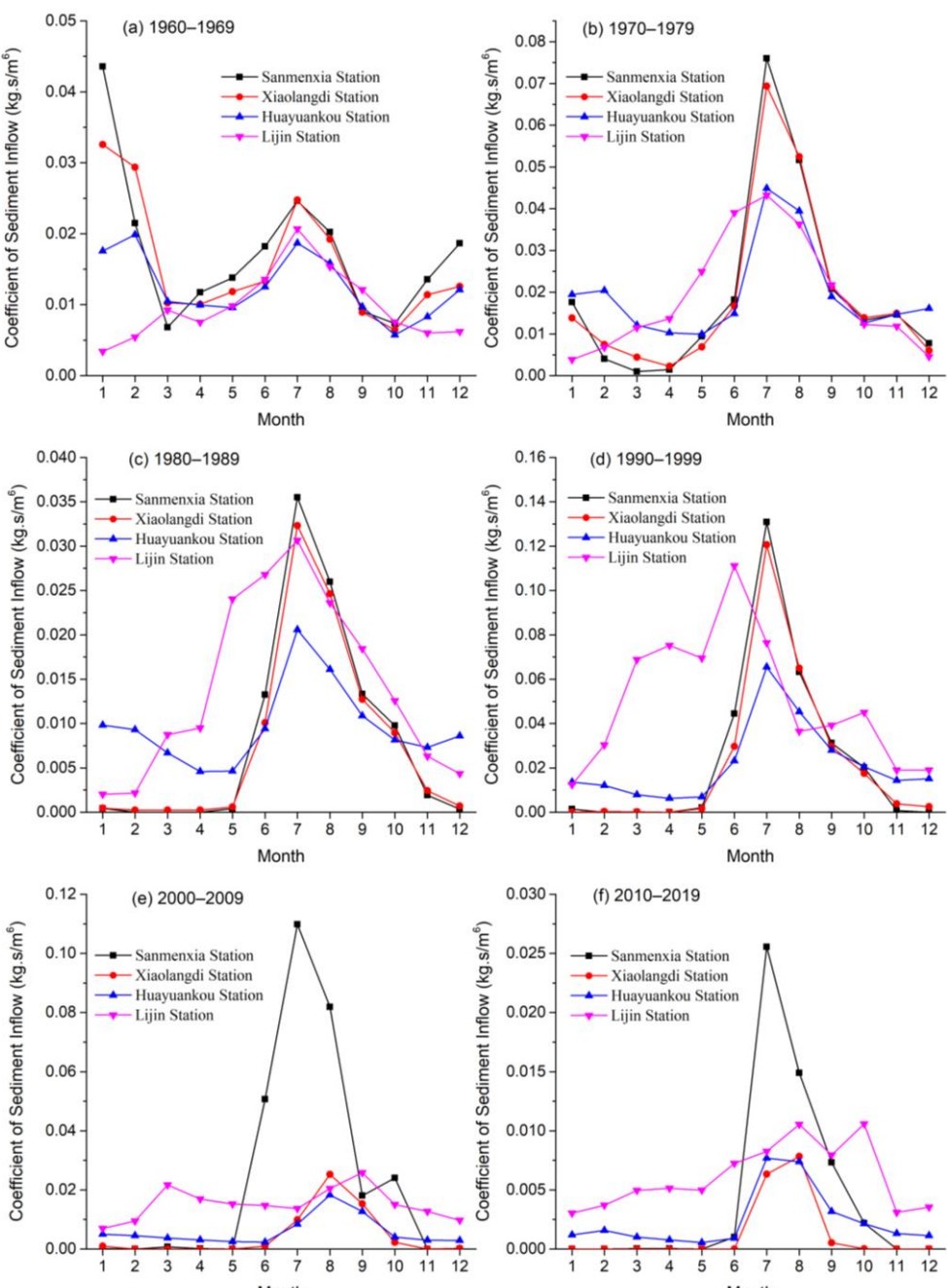

**Figure 8.** Variations of the monthly mean coefficient of sediment inflow in decadal average at the Sanmenxia, Xiaolangdi, Huayunakou, and Lijin stations of the Yellow River during the last decades.

Compared with that of the 1960s, the sediment inflow coefficient significantly increased in the summer and autumn periods in the 1970s while it significantly decreased in the winter and spring, with a larger seasonal fluctuation range (Figure 8b). This is mainly due to the fact that the sediment trap capacity of the Sanmenxia Reservoir almost disappeared and the storage capacity also was greatly consumed by sediment deposition. In the 1980s, the sediment inflow coefficient in each month was lower than that in the 1970s, and in the non-flood season, it was also lower than that in the 1960s (Figure 8c). This was mainly due to the joint operation of the Liujiaxia Reservoir and the Longyangxia Reservoir located in the upper reach of the Yellow River to reduce the flood peak. The sediment inflow coefficient in the flood seasons of the 1990s and 2000s (Figure 8d,e) was the highest and second highest among these decades. In the 2010s (Figure 8f), the sediment inflow coefficient at

the Sanmenxia station was only slightly higher in July than in the same period of the 1960s, while the other months had the smallest values among all months in all decades. At the same time, their monthly variability was also the smallest.

The distribution curve of the monthly mean sediment inflow coefficient at the Xiaolangdi Station over the four decades of the 20th century (Figure 8a–d) is basically the same as that of the Sanmenxia Station during the same period, but it is significantly different in the 2000s and 2010s, mainly manifested in the significant decrease in the sediment inflow coefficient during the flood season, especially in the 2000s, where the decrease was very significant, while the inter-monthly fluctuation range was very small (Figure 8e,f). In addition, the monthly mean sediment inflow coefficient of the Xiaolangdi Station in the 1990s was similar to that of the 1960s, except for January and February, where the curve shape was very similar.

The distribution curve of the sediment inflow coefficient at Huayuankou Station (Figure 8) was generally similar to that at the Xiaolangdi Station, with the difference that the sediment inflow coefficient during the non-flood season differentiated over time and generally shows a trend of decreasing.

For the Lijin Station, the distribution curve of the sediment inflow coefficient (Figure 8) was significantly different from that of the above three hydrological stations, which was mainly manifested in the following aspects. First, the sediment inflow coefficient at the Lijin Station in each month of the 1990s was the largest in all decades, and its monthly distribution curve presented a multi-peak pattern, Second, in the decade of the 2010s, there was a bimodal pattern of the main peak in September and the second peak in March, and at the same time, there was minimal volatility in each month in all the decades.

In summary, the sediment inflow coefficients of these hydrological stations during the flood season were the highest in the 1990s and the second highest in the 1970s. In the 2000s after the completion of the Xiaolangdi Dam, except for the Sanmenxia hydrological station, which was still very large, the sediment inflow coefficent sharply decreased at the Xiaolangdi, Huayuankou, and Lijin stations located downstream of the Xiaolangdi Dam. This is the result of a large amount of sediment retention in the Xiaolangdi Reservoir.

*3.6. Monthly Variation in Hydrologic Regime*

The monthly variation in hydrologic regime in terms of relation between monthly mean discharge and sediment concentration at the Sanmenxia Station shows that in the four decades of the twentieth century, there was an eight-shaped loop, which was composed of a counterclockwise loop for small flow stage and a clockwise loop for large flow stage (Figure 9). In both decades of this century, there was a combination of clockwise loop and linear shape.

In the 1960s, the counterclockwise loop of monthly variation in hydrologic regime at the Sanmenxia Station was significant at low flow stage (Figure 9), indicating that the sediment concentration in the river flow was relatively small at the beginning of the water level rise, and the sediment inflow was significantly lagging behind the inflow from the basin. At the end of the water level decline for the counterclockwise loop, the sediment concentration was significantly larger than that at the beginning of the water level rise, and also exhibited a sediment lag phenomenon. Therefore, the counterclockwise loop at low flow stage indicated that the sediment inflow coefficient ($\zeta_r$) at the beginning of the rising water level was less than that ($\zeta_f$) at the end of the falling water level ($\zeta_r < \zeta_f$). In the following decades, this type of counterclockwise loop at a small flow stage gradually became insignificant, and since the 2000s, it has disappeared, basically becoming a linear shape with a sediment concentration equal to zero at a small flow stage.

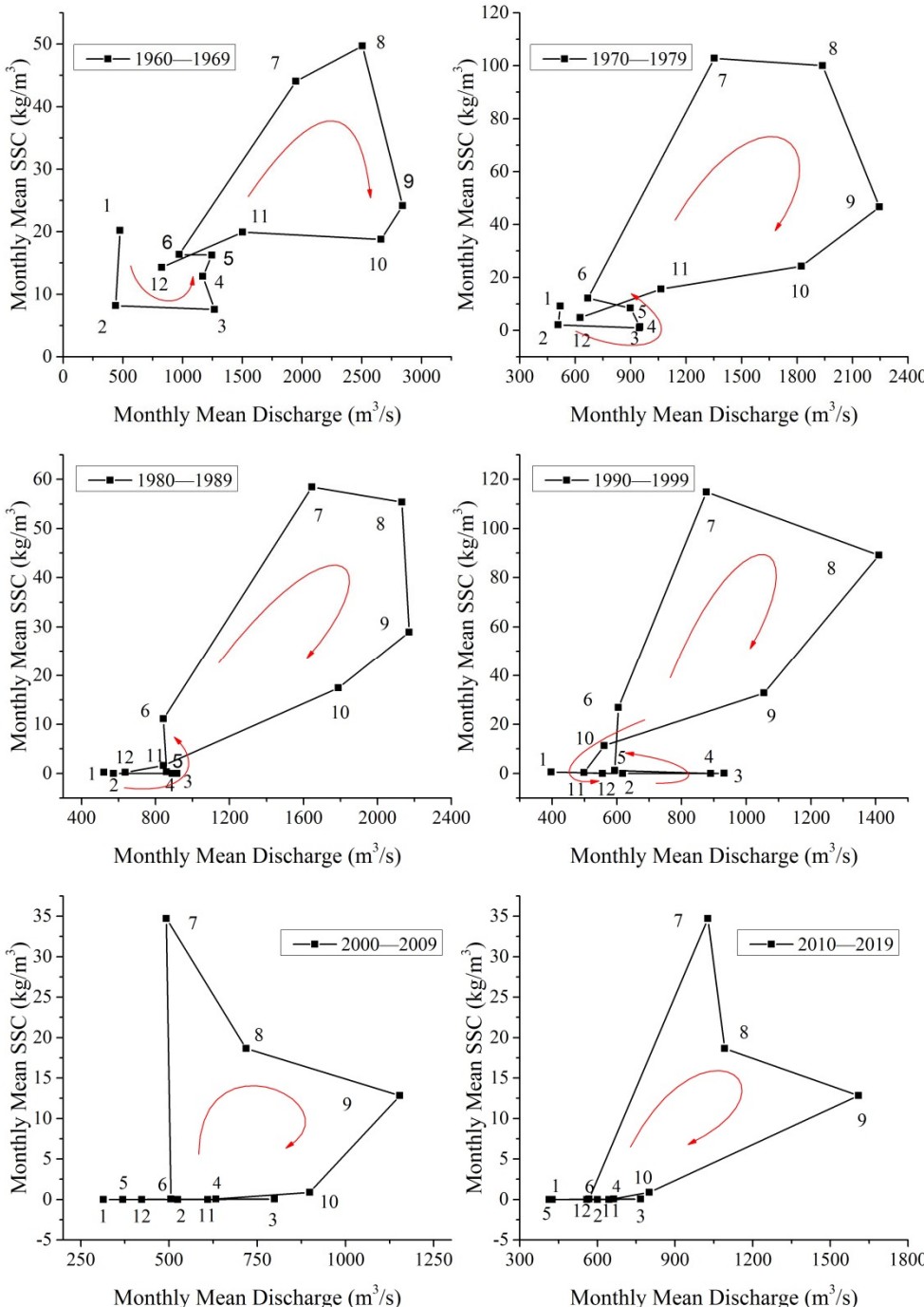

**Figure 9.** Variation in hydrologic regime in terms of process curve of monthly mean discharge and sediment concentration at the Sanmenxia gauging station in the last decades.

The clockwise loop at the Sanmenxia Station played a leading role during the six decades when the flow was in a high stage (Figure 9). Its characteristic is that the sediment concentration (sediment inflow coefficient) during the rising process of the water level is greater than that at the same water discharge during the falling process of the water level, that is, $\zeta_r > \zeta_f$. It is shown that the sediment supply from the basin surface plays a controlling role in the increasing process of water discharge and sediment concentration. During the decreasing process, the proportion of sediment concentration decrease relative to that of water discharge is greater, which is the joint result of the reduction in sediment from the basin surface and the deposition of some suspended sediment in the Sanmenxia Reservoir.

In addition, from the shape of the clockwise loop, it can be seen that in the four decades of the last century, there was a polygonal ring with convex curve connecting points, while in the two decades of this century, there was an irregular polygonal loop with a concave individual connecting point (such as the point in August). The irregular polygonal loop indicates that the sediment concentration in the water flow rapidly decreased with the increase in water discharge from July to August in the last two decades compared to that for the four decades of the last century. This phenomenon shows that the sediment retention capacity of Sanmenxia Reservoir has increased during the initial flood season (for example, from July to August) in recent years.

The monthly variation in hydrologic regime at the Huayuankou Station located at the beginning of the lower reach of the Yellow River in the different decades is shown in Figure 10. The rating curve of the hydrologic regime at this station is significantly different from that of the upstream Sanmenxia and Xiaolangdi stations. In the four decades of the last century, the rating curve at the Huayuankou Station generally presented an obvious clockwise loop, which included almost all the medium and high water stages from June to November; only when the flow was extremely small, there was a negligible linear part. In addition, this type of convex polygon clockwise loop changed into a quadrilateral loop in the 1990s, indicating that the higher monthly average sediment concentration and its significant changes mainly occur in the three months from July to September, while the sediment concentration in other months appears to be very low and almost negligible. The sediment inflow situation of the basin represented by these clockwise loops is similar to that of the Sanmenxia Station, except that the sediment concentration in the months with a falling water discharge decreases significantly as a result of partial sediment deposition in the alluvial reach upstream of the Huayuankou section.

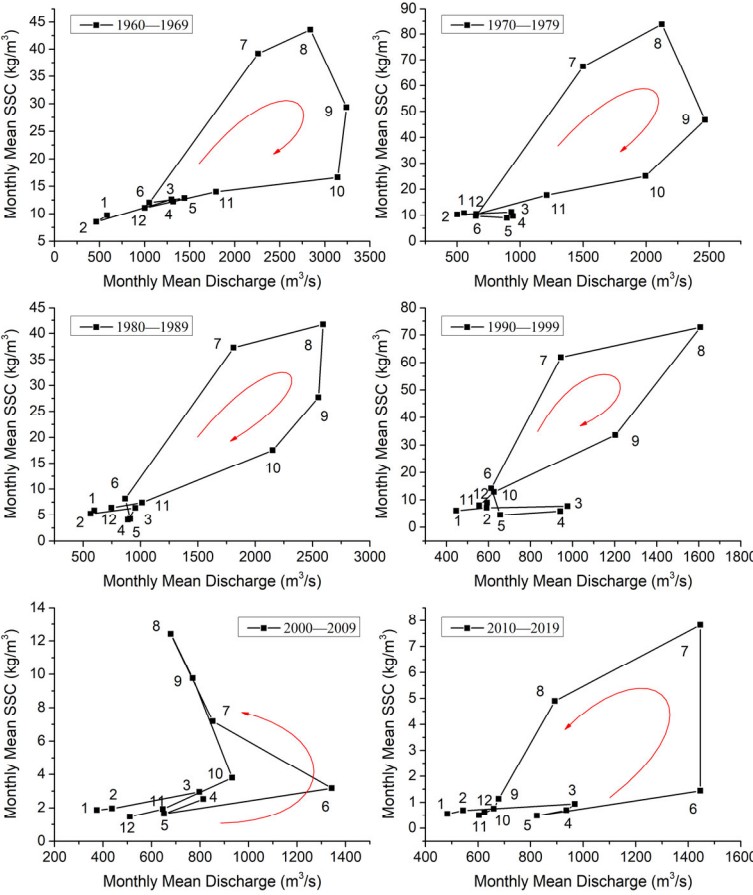

**Figure 10.** Variation in hydrologic regime in terms of process curve of monthly mean discharge and sediment concentration at the Huayuankou gauging station in the last decades.

In the 2000s, the rating curve of water discharge and sediment concentration at the station showed a combination of concave triangles and linear shapes, and the triangles belonged to a counterclockwise loop (Figure 10). This indicates that during the period from May to June, the monthly mean sediment concentration of the station increased at a low speed with the increase in the monthly mean water discharge. During the period from June to July, the water discharge was decreasing, while the sediment concentration was increasing at a low speed. For the period from July to August, the water discharge was decreasing while the sediment concentration was rapidly increasing. In the period from August to October, the water discharge increased slightly, while the sediment concentration decreased significantly. It can be seen that the monthly variation of the water sediment relationship presented more complex and diverse characteristics.

In the 2010s, there was a counterclockwise loop characterized by a convex quadrilateral shape (Figure 10). The changes in the water sediment relationship mainly occurred in July and August, indicating that during the period from June to July, the monthly mean water discharge increased at a low speed, while the monthly mean sediment concentration increased at a high speed. During the two periods from July to August and August to September, the water discharge was decreasing, while the sediment concentration was decreasing at lower and higher rates, respectively. In other months, the monthly mean sediment concentration did not change significantly with the monthly mean discharge.

## 4. Discussion

### 4.1. Ecological Protection and its Water Conservation and Sediment Reduction

The annual and monthly variations of runoff and sediment discharge in the Yellow River basin have shown a significant decrease, which is mainly affected by various human activities in the basin, while the impact of climate change is relatively limited (e.g., [5,13,21,30–33]. These human activities are mainly the continuous implementation of diversified and integrated ecological protection measures, as well as the constructions of large-scale water conservancy projects based on electricity production and water diversion that contribute to ecological improvement.

As shown above, the runoff of the Yellow River basin mainly comes from its source area (above the Tangnaihai Station, Figure 1), where the impact of human activities is relatively limited. Except for the interannual adjustment changes in runoff caused by the construction of the Longyangxia Reservoir, there is no obvious trend change in runoff. The vast majority, about 97%, of the suspended sediment transported by the Yellow River comes from the Loess Plateau [34], which occupies all of the middle reach and most of the upper reach of the Yellow River basin and is covered by loose surface material that is prone to erosion by precipitation and stream flow. Therefore, interpreting the progress of ecological protection measures in the Loess Plateau region is helpful to understand the mechanism of runoff and sediment discharge changes especially in the reach above the Xiaolangdi Station.

As is known, the Yellow River is the world's largest sediment-carrying river, and it is also a region with a very fragile ecological environment. Until the end of the last century, the water and soil loss in the middle reach was extremely serious, not only threatening the production and living activities of local people, but also causing severe siltation of the riverbed in the lower reach due to its large sediment production, which has brought great potential risks to the flood control of the vast areas on both sides of the lower reach. In order to alleviate water and sediment disasters, the Chinese government and relevant management departments have issued a number of ecological protection policies, and specialized departments have formulated various targeted ecological protection measures. Local people have actively participated in the implementation of relevant ecological protection measures.

The ecological protection work with soil and water conservation as the main purpose in the Loess Plateau has undergone several typical stages (e.g., [32,34]). The different governance stages are outlined below: small regional experiment and demonstration stage in the 1950s, comprehensive planning stage in the 1960s, comprehensive treatment

stage in the 1970s, key governance stage in the 1980s, strengthened engineering project promotion stage in the 1990s and 2000s, stage combining natural ecological restoration and water and soil conservation engineering in the 2010s. Accordingly, the total area of ecological protection such as terrace construction, grass planting, forestation, closed managed land, and flat land formed by the construction of dams for sediment retention in the Loess Plateau has continued to increase. The area of land that has been fully ecologically protected increased from 698,000 hectares in the 1960s to 1.838 million hectares in the 1970s, 3.733 million hectares in the 1980s, 6.263 million hectares in the 1990s, 9.518 million hectares in the 2000s, and over 12.459 million hectares in the 2010s. The ecological protection work in soil and water loss areas on the Loess Plateau has a very good promoting effect on water conservation and sediment reduction in the basin, and the effect is obvious (Figure 11). This has become a key factor in regulating runoff and significantly reducing suspended sediment in the middle and lower reaches of the Yellow River basin, and has also become a good example of environmental protection in river basin management.

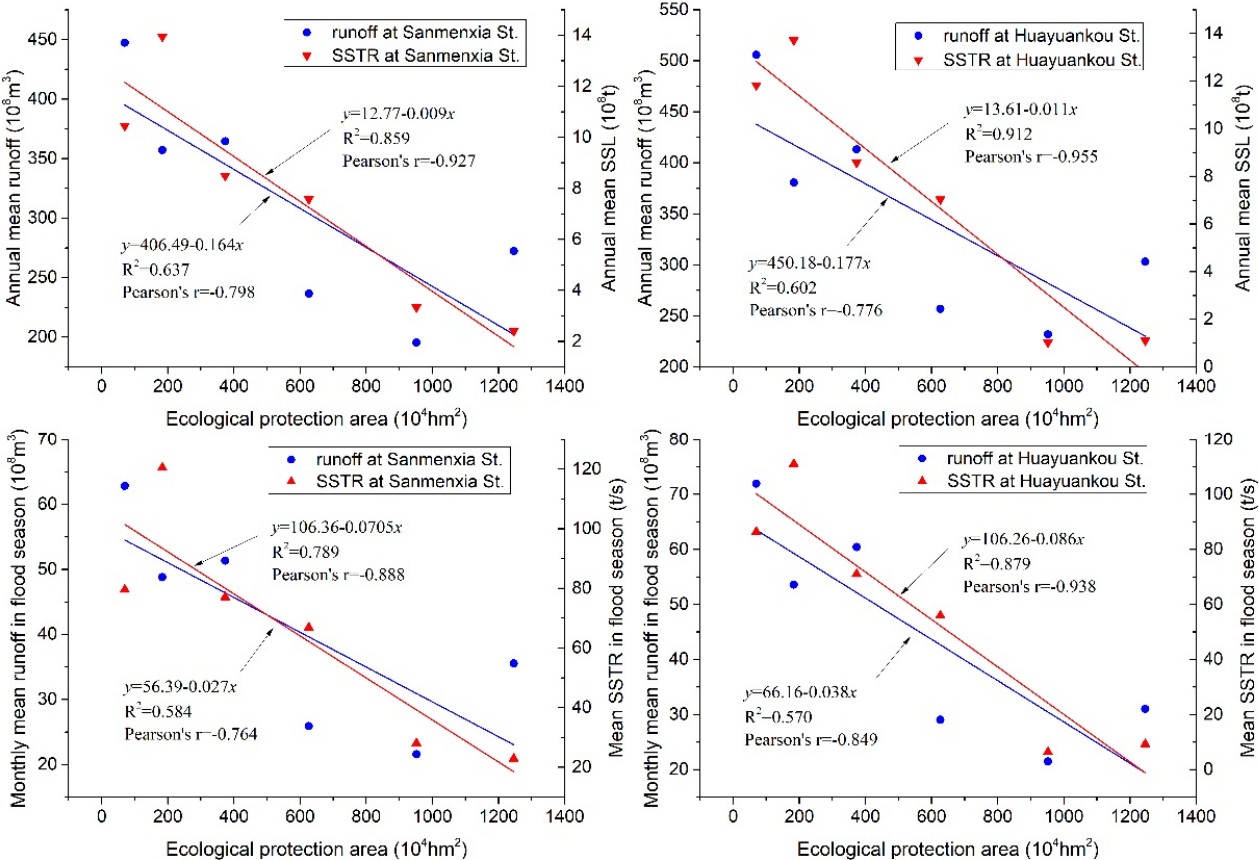

**Figure 11.** Relationships between the decadal means of annual runoff and SSTR and monthly runoff and SSTR in flood season and the ecological protection area in the Yellow River basin.

The total area of the Yellow River located in the Loess Plateau is 640,000 km$^2$, including soil loss area of 454,000 km$^2$. Since the 1970s, a series of ecological protection measures have been gradually implemented in the Loess Plateau. By 2020, the controlled soil loss area was 244,000 km$^2$, accounting for more than half of the total soil loss area in the Loess Plateau. The implementation of ecological protection resulted in mean annual decreased suspended sediment of 435 million tons for the six decades and of 800 million tons for the last two decades [35].

*4.2. Construction of Dams and Its Impact on Water and Sediment Processes*

As mentioned above, the ecological protection in the Yellow River basin, especially the Loess Plateau, has played an extremely important role in water conservation and sediment reduction in the river basin. Construction of check dams, as an important engineering measure of ecological protection on the Loess Plateau, has been developed for decades. The check dams are one of the key factors for sediment reduction in the Yellow River. The gully areas in the Loess Plateau are the main sediment-producing regions of the Yellow River. More than 90% of the sediment of the Yellow River comes from these regions. Since the 1950s, more than 59,000 check dams have been constructed in these regions [34]. These engineering measures have intercepted a large amount of sediment transported by the tributaries of the Yellow River [5,13]. Generally, these check dams undergo continuous sediment deposition and will form a series of dam lands in about 20 years, which will become fertile and leveled land in the gully region of the Loess Plateau. In addition, these dam areas have sufficient water resources, which will become a good growth base for crops, forests, and other vegetation, making an indelible contribution to water storage, sediment reduction, and ecological protection in the tributaries of the Yellow River basin.

Most areas of the Yellow River basin are located in an arid to semi-arid climate region, with precipitation mainly occurring during the flood season from July to October, showing uneven annual and interannual distribution. It is difficult for vegetation in non-flood seasons to thrive due to a lack of water resources. Therefore, the construction of large reservoirs in the main stream of the Yellow River has become a beneficial artificial water resource regulation measure for rational allocation of water resources, and can also provide clean energy (hydropower production). This not only strengthens regional ecological protection, but also reduces the consumption of non-renewable fossil energy. The large reservoirs built on the main stream of the Yellow River not only have the function of depositing sediment and accumulating water, but also significantly change the flow and sediment processes of the river.

The large reservoirs built on the main stream are located downstream of the Longyangxia, Liujiaxia, Sanmenxia, and Xiaolangdi reservoirs which were completed in 1986, 1968, 1961, and 2001, respectively. The Longyangxia and Liujiaxia reservoirs are located in the upper reach and have a limited impact on the water and sediment processes in the middle and lower reaches of the Yellow River. The construction and operation of the Sanmenxia and Xiaolangdi reservoirs, located at the end of middle reach of the Yellow River, have greatly reduced the runoff and SSL in the lower Yellow River (Figures 2–6). The annual or monthly runoff decline mainly occurred due to industrial and agricultural water use, while the SSL decrease in the mainstream was the result of sediment deposition in the large reservoirs [13]. This is consistent with other studies on the impact of reservoir construction on sediment budget of rivers in the world (e.g., [8,13,31,36–46]). Although the total storage capacity of the Sanmenxia Reservoir is 16.2 billion m$^3$, the total siltation in the reservoir amounted to 7.7 Gt during the period of 1960–1973 due to a large amount of incoming sediment [44]. Therefore, the sedimentholding capacity of the reservoir has been greatly reduced since the 1970s.

As the most downstream reservoir on the Yellow River, the Xiaolangdi Reservoir has a water area of 272 km$^2$, storage capacity of 12.65 billion m$^3$, sediment storage capacity of 7.55 billion m$^3$, water and sediment regulation storage capacity of 1.05 billion m$^3$, and long-term effective storage capacity of 5.1 billion m$^3$. Its construction and operation have changed not only the distribution of the monthly mean SSTR (Figure 7) and sediment inflow coefficient (Figure 8), but also the monthly variation pattern of discharge and sediment concentration, that is, from a clockwise loop to a counterclockwise loop (Figure 10). In the last two decades, with the implementation of measures to increase the discharge of clean water from the Xiaolangdi Reservoir before flood season in order to scour the riverbed of the lower Yellow River, annual SSL in the lower reaches of the Yellow River has increased slightly, which is also a result of changes in the reservoir operation mode. At the same time, in order to ensure sufficient ecological water demand in the lower main stream of the Yellow River, the water consumption in the middle and upper reaches of the Yellow River

has also been moderately limited, resulting in an increase in the runoff in the lower reach of the Yellow River. In any case, the adjustment of water and sediment processes caused by the construction of large reservoirs is long term and difficult to reverse.

## 5. Conclusions

Based on the daily measured water and sediment data from typical hydrological stations in the middle and lower reaches of the Yellow River during the period from 1960 to 2019, the variation trends of annual and monthly runoff and SSL were analyzed; at the same time, the monthly runoff, SSTR, sediment inflow coefficient, and hydrological regime curves in decadal average at typical hydrological stations were revealed. The main conclusions are as follows:

(1)   The interannual or monthly variation of runoff and SSL in the middle and lower reaches of the Yellow River had a significant decreasing trend in the four decades of the last century, which is mainly in response to the gradual implementation of ecological protection measures such as afforestation, grass planting, terrace construction, closure and conservation of wasteland, and check dam construction in the Yellow River basin, especially in the Loess Plateau region. In the last two decades, the runoff of the middle and lower reaches of the Yellow River has increased, while the SSL has fluctuated and increased slightly. This is a response to the implementation of new river management measures such as ensuring the ecological water demand of the lower reaches and scouring the riverbed by manually regulating water discharged from the Xiaolangdi Reservoir.

(2)   During the flood season (from July to October) in the last four decades of the last century, the monthly mean runoff and SSTR in decadal average also showed a significant trend of decreasing, especially in the 2000s and 2010s. The maximum value of the monthly mean sediment inflow coefficient in a decadal average was in the 1990s while the sub-maximum was in the 1970s for the typical hydrological stations, but the minimum and sub-minimum were in the 2000s and 2010s, respectively, except for the Sanmenxia Station. This was a comprehensive response to the environmental protection measures in the Yellow River basin, in which the construction and operation of the Xiaolangdi Reservoir played a key role.

(3)   The construction and operation of the Xiaolangdi Reservoir has changed the downstream hydrological regime in terms of the process curve of the monthly mean discharge and sediment concentration, mainly manifested in the change in curve shape of the process curve from a clockwise loop before the construction of the reservoir to a counterclockwise loop after its construction. Because the Sanmenxia Reservoir lost its sediment retention capacity early due to a large amount of sediment deposition in a few short years after its completion, its later operation will not change the water sediment relationship of downstream river.

**Author Contributions:** Conceptualization, S.W.; Formal Analysis, S.W. and X.W.; Draft Preparation, X.W.; Review and Editing, S.W. All authors have read and agreed to the published version of the manuscript.

**Funding:** This research was funded by the National Natural Science Foundation of China (Grant No. 41971004) and National Key Research and Development Program of China (Grant No. 2022YFC3203903).

**Data Availability Statement:** Data can be accessed through personal contact.

**Acknowledgments:** The authors thank the Yellow River Water Conservancy Committee for permission to access the hydrological gauging data.

**Conflicts of Interest:** The authors declare no conflict of interest.

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
