# Peer review of "Changes in Water and Sediment Processes in the Yellow River and Their Responses to Ecological Protection during the Last Six Decades"

_water, doi:10.3390/w15122285_

Round 1

Reviewer 1 Report

11: can reflect not only.

15: Change "it was analyzed" to "we analyzed" or "analysis were made on".

27: Delete the comma.

37-39: What is the meaning of “the key links to provide water and sediment fluxes on land and sea”?

48-49: Needs references for the argument "their change trend was not obvious in decades time scale".

58: Check the three figures, 19, 1.3 and 30%.

60: Delete the word mainly.

62: Delete the words “and occupies an absolute dominant position”.

72: Check the percentage 98.9%.

99 and 100: Change "is accounting" to "accounts".

108: Change tuns to tons.

160: Change Stations to stations.

164: Change is to are.

187: Change a in creasing to an increasing.

213: decades.

225: Change is to was, and add "in the recent two decades" before respectively.

232: while was a significant difference?

241: Delete the before October.

288: Delete the words "during the same period".

317: Change Spatial variation to Variations.

320: Delete the words "at the station".

355: Change "higher" to "the second highest".

372: Add was before relatively small.

374: It is not clear what "it" and "the former" refer to. What is the subject for the second clause in this sentence?

392-396: The flow increased from July to September in Figure 9. Why did you say with the decrease of river flow? Also, an explanation about the phased characteristic is needed.

402: What is variation interval?

405-406: It is hard to understand that the quadrangular loop indicates the decreasing of sediment concentration in the months when the water discharge changes significantly. In addition, it is better to specify the subject for the last clause. It should be different from the sediment concentration, the subject of the preceding clause. I guess it is the sediment discharge.  

408-411: The words after except may be rephrased to "except that the sediment concentration in months with a falling water discharge decreases significantly as a result of sediment deposition in the alluvial reach upstream of the Huayuankou section".

Figure 10: Label each chart in the figure.

450: Change with to has or is covered by.

457 Change is to was.

471: Change strengthen to strengthened.

472: natural ecological restoration.

474: Artificial forest land can be replaced by forestation.

475: Change "continues to increase" to "has continued to increase".

486: Change "annual and monthly mean runoff and SSTR in decadal average" to " the decadal means of annual runoff and SSTR and monthly runoff and SSTR in flood season".

488-494: The paragraph needs to be rephrased.

498: Delete the words "on the surface".

523: Change “The large reservoirs have bult on the main stream are downstream” to "The large reservoirs built on the main stream are".

524: Change "which have constructed" to "which were completed".

526: have a limited impact.

528: at the end.

531: was the result of sediment deposition.

532: Change "research works" to "studies".

544-545: have changed not only......, but also the monthly....

565: Change has to had.

537: Change regulated to regulating and delete the word clear.

580: Change "That vas" to "This was".

A careful language edit is necessary.

Reviewer 2 Report

This is an interesting manuscript and a very well-designed research. This paper my be accepted if the following problems can be clarified:

1. The abstract is too much, please remove some unimportant information.

2. why you choose the stations in the research? Please clarify!

3. The quality of the pictures are not well, please improve.

Reviewer 3 Report

The article entitled “Changes of water and sediment processes in the Yellow River and their responses to ecological protection during the last six decades” presents excellent scientific writing and is well structured in terms of objectives, results and discussion. In addition, it presents a good contextualization of the problems associated with the “Yellow River”.

However, it must present trend analysis to detect changes in Runoff and SSL. Visually, it is observed that there were significant changes, but they need statistical proof. Just presenting the behaviors in graphs is insufficient. I suggest using the Mann Kendall test, with 10% significance. The Mann Kendall test is based on two hypotheses: the null hypothesis (H0) which assumes that the series is stationary (no trend) and the alternative hypothesis (H1) which indicates the existence of a trend. In addition, the Mann Kendall test allows checking the direction of the trend (positive or negative) through the sign (+ or -) associated with the Z value of the test.

For the application of this test, the use of complete data series (without failures) is required, considering annual data. Thus, it will also be necessary to present the methodologies for filling gaps, in case they occur.

By assessing the amplitude of these trends (when significant), I also suggest applying Sen's robust linear regression (SEN, 1968). Sen's test is a non-parametric slope estimator based on the median and is indicated for being robust against outliers and widely used in estimating trends for climatic and hydrological variables (MUELLER et al., 2013; ZHANG et al., 2015).

If the authors implement these suggestions, I express a favorable opinion on the “acceptance” of the article.

Reviewer 4 Report

This paper is a study of long-term observations of flow, SSTR, and river changes in the lower Yellow River. The data used in the paper is rich, and the analysis is well done. 

Please show the major river structures, such as Xiaolangdi Dam, in Figure 1. Also, organize the important events in the Yellow River, such as the construction of dams, by chronology.
